# Application of pharmacogenomics to investigate adverse drug reactions to the disease-modifying treatments for multiple sclerosis: a case–control study protocol for dimethyl fumarate-induced lymphopenia

Kaarina Kowalec,[1] Elaine Kingwell,[1] Robert Carruthers,[1] Ruth Ann Marrie,[2] Sasha Bernatsky,[3] Anthony Traboulsee,[1] Colin J D Ross,[4,5] Bruce Carleton,[5,6] Helen Tremlett[1]

For numbered affiliations see end of article.

**Correspondence to**
Prof. Helen Tremlett;
helen.tremlett@ubc.ca

## ABSTRACT

**Introduction** Adverse drug reactions (ADRs) are a global public health issue. The potential for pharmacogenomic biomarkers has been demonstrated in several therapeutical areas, including HIV infection and oncology. Dimethyl fumarate (DMF) is a licensed disease-modifying therapy for the treatment of multiple sclerosis (MS). The use of DMF in MS has been associated with a severe reduction in lymphocyte counts and reports of progressive multifocal leukoencephalopathy. Here, we outline the protocol for a case–control study designed to discover genomic variants associated with DMF-induced lymphopenia. The ultimate goal is to replicate these findings and create an efficient and adaptable approach towards the identification of genomic markers that could assist in mitigating adverse drug reactions in MS.

**Methods and analysis** The population sample will comprise DMF-exposed patients with MS, with cases representing those who developed lymphopenia and controls who did not. DNA genotyping will take place using a high-throughput genome-wide array. Fine mapping and imputation will be performed to focus in on the potentially causal variants associated with lymphopenia. Multivariable logistic regression will be used to compare genotype and allele frequencies between the cases and the controls, with consideration of potential confounders. The association threshold will be set at p<1.0×10⁻⁵ for the discovery of genomic association analyses to select variants for replication.

**Ethics and dissemination** Ethics approval has been obtained from the respective research ethics board, which includes written informed consent. Findings will be disseminated widely, including at scientific conferences, via podcasts (targeted at both healthcare professionals as well as patients and the wider community), through patient engagement and other outreach community events, written lay summaries for all participants and formal publication in peer-reviewed scientific journals.

## Strengths and limitations of this study

► This study aims to minimise adverse drug reactions (ADRs), specifically dimethyl fumarate (DMF) induced lymphopenia in multiple sclerosis (MS), through the discovery of pharmacogenomic biomarkers.

► This protocol is easily adaptable to the search for genomic markers of other ADRs associated with the MS disease-modifying therapies.

► We propose to investigate a phenotype that is based on an objective laboratory measurement (the absolute lymphocyte counts), to facilitate replication in future studies.

► Results may be limited to adults with MS of specific genetic ancestries.

► The scarcity of literature on the genetic basis of drug-induced lymphopenia, combined with DMF's undefined mechanism of action, means that a hypothesis-free genome-wide approach, rather than a targeted candidate gene investigation, is preferred to identify any pharmacogenomic biomarkers of the ADR.

► Functional investigation of any identified genetic variants associated with the ADR with in vivo or in vitro models are beyond the scope of the current protocol, but will be developed if a genetic target is discovered.

## INTRODUCTION

Adverse drug reactions (ADRs) are a serious public health issue, representing the fourth to sixth leading cause of death in American hospitals,[1] and were reported to be responsible for an estimated 1 out of every 15 hospital admissions by authors of a UK-based study.[2] Over the last decade, the disease-modifying therapy options for

**BMJ**

multiple sclerosis (MS) have increased, with over 10 different products now available. However, these treatments carry some serious safety concerns. Possibly the most studied and monitored ADR to an MS disease-modifying therapy has been the fatal, or severely debilitating opportunistic infection, progressive multifocal leukoencephalopathy (PML), which is caused by reactivation of the John Cunningham virus. This ADR has primarily been associated with natalizumab, but more recently, case reports of PML with other MS disease-modifying therapies, including dimethyl fumarate (DMF), have emerged.[3][4] A severe reduction in lymphocytes (grade 3) has been associated with the subsequent occurrence of PML.[3] During the 96-week pivotal clinical trial, 1 in 20 (5%) DMF-treated patients experienced severe, 'grade 3' lymphopenia[5] (defined as absolute lymphocyte counts $<0.5$–$0.2\times10^9$/L or $<500$–$200$/mm$^3$ by the Common Terminology Criteria of Adverse Events).[6] A similar proportion of DMF-exposed patients (5.9%) experienced grade 3 lymphopenia in a subsequent post-marketing study, although this actually occurred over a shorter observation period (44 weeks), with 20% of those aged over 55 years affected.[7] The potential for a fatal or severely disabling ADR such as PML occurring during DMF exposure is a significant concern, given that DMF-exposed individuals can experience a severe reduction in their lymphocyte counts.

Currently, it is challenging to predict serious ADRs to the MS disease-modifying therapies, using clinical or demographic features. Validated pharmacogenomic biomarkers can predict an individual's ADR risk and offer an important means of facilitating appropriate drug selection, dose initiation or tailored safety monitoring.[8] Here, we focus on DMF-induced lymphopenia because of its high clinical importance in MS given its previous association with PML. To the best of our knowledge, DMF-induced lymphopenia has not been examined in the context of pharmacogenomics.

## STUDY AIMS

This protocol outlines an efficient and adaptable case–control study designed to discover pharmacogenomic markers that could be used to tailor therapy and thereby reduce the occurrence of DMF-induced lymphopenia.

Specifically, using a genome-wide association approach, we will investigate whether DMF-induced lymphopenia can be genotypically predicted.

## METHODS
### Study patients

We developed an efficient and comprehensive method to identify relevant biomarkers of drug harm through the Canadian Pharmacogenomics Network for Drug Safety, as previously described.[9] The source population for the discovery sample will comprise adult patients (≥18 years old) who were seen at the Djavad Mowafaghian Centre for Brain Health MS clinic, located at the University of British Columbia, Vancouver, Canada, diagnosed with relapsing-onset MS (based on the internationally recognised criteria current at the time of diagnosis)[10–13] and have documented exposure to DMF. Participants will be recruited 2016–2018. Cases and controls will be drawn from this source population; cases will be defined as patients who experience grade 3 lymphopenia as documented on at least one result from a laboratory test conducted during DMF exposure (table 1). Eligible controls will be patients who are 'drug tolerant' as determined by a minimum of 1-year exposure to DMF and normal lymphocyte counts as documented on all available results of laboratory test conducted during DMF exposure (table 1). Each patient's medical record will be extensively reviewed to collect clinical and demographical information including sex, date of birth, laboratory test results (date, numeric result, units and normal range assigned by the testing laboratory), prior disease-modifying therapy exposure and concomitant medication exposure (dates of initiation and cessation, drug name, dose, frequency, when available). Data will be collated by means of a structured study-specific form. A questionnaire will be administered to all cases and controls to obtain additional information not always available in the medical records: infections requiring treatment with a systemic antimicrobial agent, comorbid conditions, and patient height and weight (to calculate body mass index). These factors may represent contributing causes of lymphopenia or potential confounders of the relationship between a putative genomic marker and the ADR. A validated ADR causality assessment tool,

**Table 1** Summary of the adverse drug reaction, severity and definitions used to select the multiple sclerosis cases and drug-tolerant controls

| ADR | ADR severity* | Case definition | Control definition |
|---|---|---|---|
| DMF-induced lymphopenia | Grade 3: defined using the CTCAE as a 'severe or medically significant event' | ► Normal baseline† ALC (≤6 months prior to DMF initiation) and<br>► >1 laboratory test result with ALC<500–200/mm$^3$ or <0.5–0.2×10$^9$/L during DMF exposure[6][7] | ► Normal baseline† ALC (≤6 months prior to DMF initiation) and<br>► Exposure to DMF for ≥1 year[5] and<br>► ≥1 ALC test performed during treatment, with all results within the normal range† |

*Graded by the Common Terminology Criteria for Adverse Events (CTCAE) (V.4.03).[6]
†Normal range defined by the testing laboratory.
ADR, adverse drug reaction; ALC, absolute lymphocyte counts; DMF, dimethyl fumarate.

the Naranjo scale,[14] will be employed to assist with the exclusion of competing aetiologies. A 1-year time frame is anticipated for patient recruitment for inclusion in these 'discovery stage' analyses. Following the discovery of any genomic variants that reach the pre-determined statistical threshold of association with the ADR, a replication cohort will be identified and recruited.

For the discovery stage, we require 156 patients (52 cases and 104 controls) to have sufficient power (80%) to identify a genomic variant (minor allele frequency >0.15) with a clinically significant effect size (odds ratio >5.0) and application of the National Human Genome Research Institute genome-wide association study catalogue threshold of $p<1.0\times10^{-5}$.[15] The sample size estimate is dependent on the expected minor allele frequency of the genomic variant. Given that there are no prior data from previous pharmacogenomic studies with DMF, a minor allele frequency of 0.15 was used, as estimated by a previous pharmacogenomic study of ADRs in MS.[16] Sample size estimation was performed using Quanto (V.1.2.4).[17]

### Ethics, consent and permissions

The University of British Columbia Clinical Research Ethics Board and the Vancouver Coastal Health Research Institute have approved this study protocol (H16-01927), which includes written informed consent from all participants.

### Collection of saliva, genotyping and fine mapping

A saliva sample will be obtained from patients, either during their clinic visit or collected at home and returned via mail, with Oragene collection tubes (DNA Genotek). DNA will be extracted from saliva with the QIAmp DNA purification system (Qiagen) and subsequently quantified using the Quanti-iT PicoGreen assay (Invitrogen). DNA samples will be stored at −20°C until all samples have been collected.

Samples will be processed with the Illumina Tecan Freedom EVO 150 and scanned on the Illumina HiScan System (Illumina). Every set of 96 samples will include a negative (1x Tris-EDTA buffer) and positive control. Samples will be genotyped using a genome-wide genotyping array. We propose a hypothesis-free, genome-wide approach for the identification of potential pharmacogenomic biomarkers related to this ADR. Rationale for this approach stems from the dearth of literature on the genetic basis of drug-induced lymphopenia, including no relevant pharmacogenomic studies with DMF. Moreover, as the exact mechanism of DMF-lymphopenia is undefined, it is impractical to predict which genes are involved in this ADR. In the event that validated pharmacogenomic markers of drug-induced lymphopenia are available at the time of genotyping, we will consider these in the current study.

For quality control purposes, genomic variants will be excluded with call rates <90%, a minor allele frequency <1% or those deviating from the Hardy-Weinberg equilibrium genotype distribution ($p<1.0\times10^{-5}$ in controls). Individual samples will also be excluded with call rates <95%, and those cryptically related to other samples, or if the individual's sex according to the genotyping is discordant with that recorded in the medical records.

We will perform linkage disequilibrium analyses ($r^2$ and D') using the 1000 Genomes CEU population (Utah residents with Northern and Western European ancestry) (http://www.1000genomes.org) and other populations, as appropriate. Haplotypes and haplotype blocks will be calculated using CEU 1000 Genomes population as described.[18] Haplotype analyses will be implemented using Golden Helix SNP and Variation Suite (V.8.4; Bozeman).

Fine mapping and imputation will be performed to uncover putative causal variants. We will perform genetic fine-mapping analysis by genotype imputation of additional single nucleotide polymorphisms (SNPs) not present on the genotyping platform using BEAGLE V.4.0 with LD and haplotype information from 1000 Genomes (phase 3) as the reference population.[19 20] We will impute additional variants on the chromosome region (±500 kb) containing any significant variants identified during the discovery phase. Any significant variants within the imputed region will be assessed by means of the Combined Annotation Dependent Depletion framework[21] and corroboratory evidence regarding the function of any associated variants will be explored from the literature. The quality control metric (BEAGLE allelic $R^2$) will be calculated for all imputed SNPs and will include SNPs with an $R^2 \geq 0.5$ for analysis.

Given the association between specific human leucocyte antigen (HLA) alleles and a first-line disease-modifying therapy for MS, interferon-beta,[22–24] we will also investigate whether an association between DMF-induced lymphopenia and specific HLA alleles exist. We will directly type *HLA-DRB1\*15:01*, in addition to using the HLA-specific imputation programme, SNP2HLA,[25] to determine the specific HLA alleles of the individuals.

The genotyping calls of any significant variants from the genome-wide array, or those prioritised by fine mapping, will be validated by TaqMan genotyping assay (ThermoFisher Scientific).

### Replication of findings

We will seek to replicate any variants that reach the pre-determined threshold in this initial cohort in an independent replication cohort within the Canadian Network of MS Clinics (www.cnmsc.ca). All participant inclusion and exclusion criteria will match that of the criteria used in the discovery cohort.

### Data management

The principal investigator (HT) and the study team will manage all the patient data for this study. All data will be stored on a secure server, on a password-protected computer, located at the University of British Columbia, Faculty of Medicine.

## Statistical analyses

Clinical and demographical factors will be compared between cases and controls. Categorical variables will be summarised by frequency (per cent) and analysed using either parametric ($\chi^2$ test) or non-parametric tests (Fisher's exact test). Continuous variables will be summarised by describing mean (SD) or median (IQR) and analysed using the appropriate parametric (unpaired Student t-test) or non-parametric test (Mann-Whitney U test). Genomic ancestry will be ascertained by means of principal components analysis (EIGENSTRAT method)[26] using the genotyping data and the 1000 Genomes Project reference data.[19]

Differences in genotype and allele frequencies between cases and controls will be tested using logistic regression, with DMF-induced lymphopenia as the outcome. The logistic regression model will use an additive genetic model to identify associations since this model has been used previously to detect significant genetic differences associated with cases of ADRs.[27] [28] In addition, the additive model is the most common genetic association test in which the underlying genetic model is unknown.[29] The regression analyses will be adjusted for any relevant characteristics, such as sex, age, genetic ancestry or previous disease-modifying therapy exposure.[7] To examine evidence of additional genetic associations with the ADR in the imputed region, the same analysis and models used in the initial analysis will be applied.

For the genomic association analyses, the threshold to indicate variants that will require replication will be $p < 1.0 \times 10^{-5}$.[15] During the replication stage of the genomic analysis, we will employ a Bonferroni-corrected $p < 0.05/n$ (n = number of significant SNPs from the discovery phase). We will include any genetic variants reaching the predetermined significance threshold into the logistic regression model and adjust for potential confounders. Golden Helix SNP and Variation Suite (V.8.4; Bozeman) and IBM SPSS (V.23) will be used to conduct the statistical analyses. Visual plots (Manhattan and regional association plots) will be generated using Golden Helix and LocusZoom.[30]

## Dissemination

Findings will be disseminated via various avenues, including at scientific conferences, via podcasts (targeted to healthcare professionals, patients and the general public), through patient engagement and other outreach events, written lay summaries for all participants and formal publication in peer-reviewed scientific journals.

## DISCUSSION

ADRs present a significant burden to global healthcare systems. In the UK alone, ADRs were estimated to cost the National Health Service £466 million (US$847 million) in 2002.[2] A more recent Swedish study reported the overall annual direct cost of adverse drug events as US$21 million per 100 000 habitants, including the costs of diagnosing, monitoring or treating the events.[31] Neither the financial costs nor the morbidity and mortality rates associated with ADRs to the MS disease-modifying therapies are fully known as yet; this is despite the ubiquitous use of these therapies. Given the clinical importance of ADRs, it is highly desirable to minimise their occurrence. One approach to minimising ADRs is through the discovery and implementation of pharmacogenomic biomarkers; an example of the utility of this approach is demonstrated by the now commonly accepted clinical practice of testing for the *HLA-B\*57:01* allele in HIV-infected persons to prevent abacavir-induced hypersensitivity.[32]

Our protocol, which is aimed here at discovering biomarkers associated with DMF-lymphopenia, is readily adaptable to the search for genomic markers of other ADRs associated with the MS disease-modifying therapies (eg, alemtuzumab-induced immune thrombocytopenia). There are several key considerations when conducting pharmacogenomic studies of serious ADRs.[33] Arguably one of the most important is the thorough phenotyping of cases and controls, as poor characterisation contributes to the lack of replication between studies.[8] We have proposed to investigate a phenotype that is based on an objective laboratory measurement (the absolute lymphocyte counts); this may facilitate replication in future studies. A genome-wide, hypothesis-free method, as proposed here, could be considered an ideal approach to identify pharmacogenomic biomarkers associated with an ADR because the mechanism of action of DMF is unclear,[34] making it difficult to pre-specify or hypothesise which genes might be involved in the ADR. Other important considerations when designing pharmacogenomic studies of serious ADRs include the appropriate sourcing of a second cohort to replicate the findings from the discovery stage; a multi-centre collaboration is being sought through the Canadian Network of MS clinics (http://www.cnmsc.ca); this has been successfully achieved in a prior study.[16] Such collaboration can also serve to demonstrate the generalizability of findings across multiple sites.

The disease-modifying therapy options that are available for people with MS have increased substantially over the last decade. This rise in therapeutic choices has provided many benefits, including non-injectable options and, for the newer therapies, increased efficacy over the previous MS disease-modifying therapies.[5] [35] [36] However, serious ADRs are associated with these newer disease-modifying therapies, and there are currently few means of predicting whom they might affect. If pharmacogenomic biomarkers can be identified by the methods proposed here, this would offer a 'precision medicine' approach to managing MS drug treatments by facilitating choice of optimal disease-modifying therapy or by tailoring safety monitoring. Ultimately,

the incorporation of pharmacogenomic biomarkers into the complex therapeutical decision-making process would benefit MS patients and healthcare providers.

**Author affiliations**
[1]Faculty of Medicine, Division of Neurology and Djavad Mowafaghian Centre for Brain Health, University of British Columbia, Vancover, British Columbia, Canada
[2]Departments of Internal Medicine and Community Health Sciences, Max Rady College of Medicine, Rady Faculty of Health Sciences, University of Manitoba, Winnipeg, Canada
[3]Division of Rheumatology, McGill University, Montreal, Quebec, Canada
[4]Faculty of Pharmaceutical Sciences, University of British Columbia, Vancouver, British Columbia, Canada
[5]B.C. Childrens Hospital Research Institute, Vancouver, Canada
[6]Faculty of Medicine, Department of Pediatrics, University of British Columbia, Vancouver, Canada

**Contributors** KK, EK and HT contributed to the concept and design of the study, and drafting of the article. All authors reviewed, commented and approved the final manuscript.

**Funding** This work was supported by the Canadian Institutes of Health Research (CIHR) – Drug Safety and Effectiveness Network (British Columbia site PI: Tremlett).

**Competing interests** AT has received grant support from Hoffman la Roche and Sanofi Genzyme; steering committee membership for Hoffman la Roche; consultancy for Biogen, Chugai, EMD Serono, Hoffman la Roche, Medimmune, Sanofi Genzyme and Teva Neuroscience. RAM has conducted clinical trials for Sanofi-Aventis and receives research funding from CIHR, the National MS Society, the MS Society of Canada, the MS Scientific Research Foundation, Research Manitoba and the Waugh Family Chair in Multiple Sclerosis. CJDR receives funding support from the Canadian Institutes of Health Research, Canadian Foundation for Innovation, Canadian Hearing Foundation, BC Children's Hospital Foundation, BC Children's Hospital Research Canadian Gene Cure Foundation, Teva Pharmaceutical Industries Inc., Genome BC and the CIHR Drug Safety & Effectiveness Network. BC currently receives research funding from the Canadian Institutes of Health Research, Genome Canada, Genome British Columbia and BC Children's Hospital Research (Vancouver, Canada) and has previously held matching funds support for Genome Canada funding from Pfizer Canada (unrestricted). HT has received speaker honoraria and/or travel expenses to attend conferences in the last 5 years from the Consortium of MS Centres (2013), National MS Society (2012, 2014, 2016), ECTRIMS (2012–2016), the Chesapeake Health Education Program, US Veterans Affairs (2012), Novartis Canada (2012), Biogen Idec (2014) and AAN (2013–2016). All speaker honoraria are either declined or donated to an MS charity or to an unrestricted grant for use by her research group.

**Ethics approval** The University of British Columbia Clinical Research Ethics Board and the Vancouver Coastal Health Research Institute.

**Provenance and peer review** Not commissioned; externally peer reviewed.

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
