## [Reviewer comments · BMJ Open]

ARTICLE DETAILS

TITLE (PROVISIONAL)	Application of pharmacogenomics to investigate adverse drug reactions to the disease-modifying treatments for multiple sclerosis: a case-control study protocol for dimethyl fumarate induced lymphopenia
AUTHORS	Kowalec, Kaarina; Kingwell, Elaine; Carruthers, Robert; Marrie, Ruth; Bernatsky, Sasha; Traboulsee, Anthony; Ross, Colin; Carleton, Bruce; Tremlett, Helen

VERSION 1 - REVIEW

REVIEWER	William Sheremata Miller School of Medicine United States
REVIEW RETURNED	26-Feb-2017

GENERAL COMMENTS	The authors present a protocol for a pharmacogenomic case control using GWAS to investigate lymphopenia as an adverse drug reaction to dimethyl fumarate. The protocol is clearly written with clear objectives. This is a "discovery" phase of investigation where data from a relatively small number of subjects becomes the basis for further investigation that is alluded to in this protocol. This could be more clearly stated. The importance of lymphopenia as a potential serious adverse experience with dimethyl fumarate is described. The statement that a 5% incidence of lymphopenia is relatively high proportion of the population may be open to question. The degree of lymphopenia (the phenotype) required for study entry has been clearly defined in terms of absolute lymphocyte accounts. One might question why the study was designed to include "a relatively large" number of subjects for the discovery portion of the study, a preliminary non-hypothesis driven study.
---

REVIEWER	Hans-Peter Hartung Department of Neurology Heinrich-Heine-University Düsseldorf, Germany
REVIEW RETURNED	10-Apr-2017

GENERAL COMMENTS	Dimethylfumarate has been linked to lymphocytopenia, and this in turn to the development of PML. Decreases in lymphocyte counts on DMF treatment is therefore a significant concern of relevance in the management of these patients. The authors report an approach and a protocol by which they want to conduct a case-control study with the aim of identifying pharmacogenomic markers that would aid in the prediction of lymphopenia. Methods are well described, statistical power calculation appears right and the study could yield important outcomes.
--

	More generally, this would delineate a way to find biomarkers helpful in mitigating serious adverse events associated with DMTs in MS.
--	--

VERSION 1 – AUTHOR RESPONSE

Reviewer: 1
 William Sheremata
 Miller School of Medicine, United States

Response: We greatly appreciate Dr. Sheremata’s helpful comments. In response to the first comment, we have added the following revisions to the manuscript to clarify that we are indeed describing the discovery phase of investigation with a plan to replicate any findings:

- Abstract (Page 2, Lines 16-22): “Here, we outline the protocol for a case-control study designed to discover genomic variants associated with dimethyl fumarate-induced lymphopenia. The ultimate goal is to replicate these findings and to create an efficient and adaptable approach towards the identification of genomic markers that could assist in mitigating adverse drug reactions in MS.”
- Main Body (Page 6, Lines 26-29): “A one-year time frame is anticipated for patient recruitment for inclusion in these ‘discovery stage’ analyses. Following the discovery of any genomic variants that reach the pre-determined statistical threshold of association with the ADR, a replication cohort will be identified and recruited.”
- Main Body (Page 6, Line 53 to Page 7, Line 3): “For the discovery stage, we require 156 patients (52 cases and 104 controls) to have sufficient power (80%) to identify a genomic variant (minor allele frequency > 0.15) with a clinically significant effect size (odds ratio > 5.0) and application of the National Human Genome Research Institute genome wide association study catalog threshold of $p < 1.0 \times 10^{-5}$.”
- Main Body (Page 9, Lines 38-41): “We will seek to replicate any variants that reach the pre-determined threshold in this initial cohort in an independent replication cohort within the Canadian Network of MS Clinics (www.cnmsc.ca).”
- Main Body (Page 12, Lines 28-31): “Other important considerations when designing pharmacogenomic studies of serious adverse drug reactions include the appropriate sourcing of a second cohort to replicate the findings from the discovery stage;”

We agree with the reviewer’s second comment that a 5% incidence of lymphopenia is not necessarily high, but the potential importance of this depends on the context. We have modified this paragraph accordingly by removing the statement:

‘Given the relatively high proportion of DMF exposed individuals experiencing reductions in their lymphocyte counts, the potential for PML occurring during DMF exposure is a significant concern.’

And replacing with modified text highlighting the importance and rationale for focusing on lymphopenia

- Main Body (Page 4, Lines 26-45): “A severe reduction in lymphocytes (Grade 3) has been associated with the subsequent occurrence of PML (3). During the 96-week pivotal clinical trial, 1 in 20 (5%) DMF-treated patients experienced severe, ‘Grade 3’ lymphopenia (5) (defined as absolute lymphocyte counts $< 0.5-0.2 \times 10^9/L$ or $< 500-200/mm^3$ by the Common Terminology Criteria of Adverse Events (6)). A similar proportion of DMF-exposed patients (5.9%) experienced Grade 3 lymphopenia in a subsequent post-marketing study, although this actually occurred over a shorter observation period (44 weeks), with 20% of those aged over 55 years affected (7). The potential for a fatal or severely disabling ADR such as PML occurring during DMF exposure is a significant concern, given that DMF exposed individuals can experience a severe reduction in their lymphocyte counts.

In response to the last comment from Dr. Sheremata. We are indeed conducting a hypothesis-free genome-wide approach, rather than a targeted candidate gene investigation. We agree that we have included a relatively large number of subjects at the discovery stage. This is to ensure detection of genomic variants with a minor allele frequency of 0.15. We have added the following statement to the manuscript to clarify this:

- Main Body (Page 7, Lines 5-14): “The sample size estimate is dependent upon the expected minor allele frequency of the genomic variant. Given that there is no prior data from previous pharmacogenomic studies with DMF, a minor allele frequency of 0.15 was used, as estimated by a previous pharmacogenomic study of ADRs in MS (34).”

Reviewer: 2

Hans-Peter Hartung

Department of Neurology, Heinrich-Heine-University Düsseldorf, Germany

Response: We thank Dr. Hartung for his supportive and encouraging comments.